# In-Field Comparative Study of Landraces vs. Modern Wheat Genotypes under a Mediterranean Climate

**DOI:** 10.3390/plants10122612

**Published:** 2021-11-28

**Authors:** Sivan Frankin, Rajib Roychowdhury, Kamal Nashef, Shahal Abbo, David J. Bonfil, Roi Ben-David

**Affiliations:** 1Department of Vegetable and Field Crops, Institute of Plant Sciences, Agricultural Research Organization–Volcani Institute, Rishon LeZion 7528809, Israel; sfrankin@volcani.agri.gov.il (S.F.); rajibroychowdhury86@gmail.com (R.R.); kamal@volcani.agri.gov.il (K.N.); 2The Robert H. Smith Institute of Plant Sciences and Genetics in Agriculture, The Hebrew University of Jerusalem, Rehovot 7628604, Israel; shahal.abbo@mail.huji.ac.il; 3Department of Vegetable and Field Crop Research, Agricultural Research Organization, Gilat Research Center, MP Negev 8531100, Israel; bonfil@volcani.agri.gov.il

**Keywords:** wheat landraces, durum wheat, bread wheat, semi-arid climate, water-stress, yield stability

## Abstract

The Near East climate ranges from arid to a Mediterranean, under which local wheat landraces have been grown for over millennia, assumingly accumulating a unique repertoire of genetic adaptations. In the current study, we subjected a subset of the Israeli Palestinian Landraces (IPLR) collection (*n* = 19: durum and bread wheat landraces, modern wheat cultivars, and landraces mixtures) to full-field evaluation. The multifield experiment included a semiarid site (2018–2019, 2019–2020) under low (L) and high (H) supplementary irrigation, and a Mediterranean site (2019–2020). Water availability had a major impact on crop performance. This was reflected in a strong discrimination between environments for biomass productivity and yield components. Compared to landraces, modern cultivars exhibited significantly higher grain yield (GY) across environments (+102%) reflecting the effect of the Green Revolution. However, under the Gilat19 (L) environment, this productivity gap was significantly reduced (only +39%). Five excelling landraces and the durum mix exhibited good agronomic potential across all trails. This was expressed in relatively high GY (2.3–2.85 t ha^−1^), early phenology (86–96 days to heading) and lodging resistance. Given the growing interest of stakeholders and consumers, these might be considered future candidates for the local artisanal wheat grain market. Yet, this step should be taken only after establishing an adjustable field management protocol.

## 1. Introduction

Wheat is a staple cereal crop that is cultivated on more than 200 million hectares worldwide in a wide range of farmland environments [1]. The Near East climate ranges from arid (<200 mm annual rainfall) to Mediterranean (>500 mm annual rainfall), under which wheat yield is limited by both rainfall and hot temperatures. Most of the world’s spring-wheat production is grown under water-limited conditions [2]. Developing crop varieties with high yield potentials through identifying drought tolerance mechanisms is important for increasing yields in dry areas [3,4]. Yield vulnerability under rainfed conditions has been examined in various studies in the Middle East, e.g., [5,6,7,8,9], including applications of different management practices in order to minimize yield reduction in dryland farming [10,11]. To cope with climate change and meet the needs of new varieties in marginal areas, researchers and breeders are constantly looking for new sources of genetic variability. Wheat landraces represent an invaluable genetic resource for breeding under changing environments [12] and often harbor rich genetic diversity [13]. According to Denčić et al. [14], there is a general agreement that modern high-yielding wheat cultivars (often with short stature) are more adapted to favorable growing conditions, whereas old cultivars and landraces have more stable yield under drought stress conditions due to their rich and complex ancestry representing vast variation in response to many diverse stresses, deriving many sustainable traits from their heritage [15]. Landraces are also important sources of beneficial alleles for GY and quality in low-producing environments. The warming and aridification in the Near and Middle East associated with climate change may have consequences for agriculture through meteorological conditions that influence crop growth and yield [16]. Durum wheat (*Triticum durum* Desf.) land area is predicted to decrease by 19% till 2050 and by 48% at the end of the century as a result of climate change [17]. Without breakthroughs for adaptation, rising temperatures in the hottest wheat-producing environments could reduce domestic production, increase dependency on imports, and threaten food security for millions [1]. During the past few decades, interest in landrace conservation has been growing, with much research recently focusing on local adaptation, stress tolerance, yield stability, and health benefits. Recently, the Israeli Palestinian Landraces (IPLR) collection was constructed, representing a wide and unique untapped genetic resource [13]. In the present study, we examined the field performance of a representative subset of landraces from the IPLR collection in comparison to modern cultivars. Those were tested under different environments with distinct levels of water supply. The specific goals of the current study were to perform a comparative agronomical multisite field study of a diverse panel of durum and bread wheat (*Triticum aestivum* L.) landraces vs. modern hard spring semi-dwarf cultivars, and to examine the feasibility of cultivating landraces for the local market.

## 2. Results

In general, landraces of bread and durum wheat present a wide phenotypic range in vegetative- and reproductive-yield-related traits. Principal components analysis (PCA) accounted for 74.6% of the field phenotypic variance for the 2018–2019 and 2019–2020 seasons (Figure 1). In both seasons, a clear PC2-based discrimination between modern cultivars and landraces was evident irrespective of crop species (durum or bread wheat) and across growing environments (separation marked by green dashed line with only two cultivars as outliers in Figure 1). Discrimination was also evident based on growing environment: in 2018–2019, this clear separation was mainly driven by additional supplementary irrigation (Δ105 mm of additional irrigation in the high (H) compared to the low (L) water treatments) and is evident both across landraces and modern cultivar groups (yellow dashed line). The most arid environment [Gilat19 (L) with 262 mm] is clustered in the third quadrant of the PCA chart toward canopy temperature (CT, measured 87–95 DfE) expressing, as expected, high CT under water-limited conditions (Figure 1). Except for Gilat19 (L), the modern cultivars assemble closely in the second quadrant with positive loading of TKW and GY. The landraces are widely scattered due to water availability, where (L) negatively affected reproductive traits (e.g., SpN, SpW, and TKW) (Figure 1). In 2019–2020, in addition to the clear discrimination between landraces and modern cultivars, there is no clear separation between environments. This might be explained by the relatively high precipitation in 2019–2020 in Gilat that minimized the differences between supplementary irrigation treatments (H vs. L) to a mere 50 mm. The high precipitation in Rishon LeZion20 (704 mm) provided a little advantage for modern cultivars in GY but not in TKW. Landraces in the wetter environments, Gilat20 (H) and Rishon LeZion20, exhibited high DM_M, SpN, SpW, and PH (Figure 1).

We divided the panel into four genotypic groups: modern bread wheat, modern durum wheat, bread wheat landraces and durum landraces. Analyses of variance (ANOVA) enable to test the significance of the various factors: genotypic group, environments and their interaction. The genotype factor was significant for GDDtH, HI, GPC and for all the yield components traits; SpN, GN, TKW and GY (Table 1). The effect of the environment was significant in all traits except GDDtH. ANOVA showed significant interactions between the genotypes and environments in PH, GN and GY (Table 1). The linear model was able to explain most of the phenotypic variability [R^2^ value ranging from 0.5 to 0.9 (excluding GDDtH); Table 1].

Modern durum cultivars maintained higher yield compared to durum landraces in all the environments except for one outlier, cv. Solet, exhibiting low GY in the two most extreme environments [Gilat19 (L) and Rishon LeZion20] (Figure 2). Interestingly, an inverse correlation between yield and late ripening was observed only in landraces but not among modern cultivars. Phenology expressed by GDDtH was negatively associated with GY (r = −0.44) among durum landraces. The longer heading period (1475–1507 GDDtH, equivalent to 107–114 DtH), decreased the overall mean yield of durum landraces to 2.4 t ha^−1^ across the five environments. Interestingly, among the early modern durum cultivar, which averaged at 4.2 t ha^−1^, GDDtH was positively associated (r = 0.73) with GY (Figure 2). Bread wheat had the same pattern vis-a-vis phenology, with modern wheat yielded an average of 4.9 t ha^−1^ and landraces average yield decreased to 2.1 t ha^−1^ (data not shown).

The GY, TKW, and DM_M responses of the four genotypic groups (modern bread wheat, modern durum wheat, bread wheat landraces, and durum landraces) under each of the five environments are presented in Figure 3a–c. A positive GY response to water accessibility is evident in the two groups of modern cultivars (Figure 3a). Landraces’ GY productivity, however, show a different trend, where durum and bread wheat landraces produced 2.4 and 2.1 t ha^−1^ range across all environments, respectively, with a noticeable response to water availability (Figure 3a). However, there is an exception across all four groups, where the high precipitation at Rishon LeZion20 (704 mm) resulted in a GY reduction, expressed also in TKW reduction (excluding modern bread wheat under the Mediterranean environment) (Figure 3b). This probably occurred due to excess soil moisture during vegetative growth (138.3 and 356.9 mm during December 2019 and January 2020, respectively; Figure A1) that consequently promoted tillering, high biomass production, and lodging (Figure 3c). Similarly, a negative effect of high maximum temperature during the grain filling stage (29.6, 33.7 and 41.4 °C) was recorded in March, April, and May 2020, respectively (Figure A1), which might have also resulted in lower TKW and eventually lower GY (Figure 3b). Notably, the TKW of the modern durum group was significantly higher than that of the durum landraces except for the Rishon LeZion20 growing environment. In contrast, the TKW of modern bread wheat did not differ from bread wheat landraces in any of the environments (Figure 3b). In addition, crop productivity can also be expressed in DM biomass at maturity. As can be observed, all four genotypic groups exhibited a constant increase in DM productivity with increasing water availability (Figure 3c). However, similar to GY, under the Mediterranean environment (Rishon LeZion20) DM_M mean production was slightly lower among the landraces groups (Figure 3c).

Lodging was rated three times during the crop vegetative and reproductive stages (Table 2). Since the lodging situation worsened as the season progressed for most of the genotypes, we chose the latest scoring date for comparing the genotypes across environments (Figure 4). Lodging severity clearly discriminated between lines that tended to lodge, those that did not lodge, and those that exhibited intermediate lodging in any of the environments. Notably, in the driest environment (Gilat19, 262 mm), none of the genotypes lodged (except Karun, with a lodging score = 1). Some of the genotypes had severe lodging in most of the environments, probably the result of culm length, plant height, and late phenology. Plant height was highly correlated with lodging among durum and bread wheat landraces (r = 0.66 and 0.53, respectively; Figure 4b). Durum landraces’ lodging rate was also significantly correlated with GDDtH (r = 0.38; Figure 3c). In addition, lodging negatively correlated with yield components in durum (TKW; r = −0.7) and bread wheat landraces (HW, r = −0.49; and TKW, r = −0.033; data not shown). Conversely, as expected, the semidwarf modern cultivars of both durum and bread wheat showed lodging resistance in all the environments (except cv. Ruta with a lodging score = 1.5) in Rishon LeZion20 (704 mm) (Figure 4a).

The superiority index (Pi) was used to assess cultivar general superiority across the various environmental field experiments. The Pi was calculated for GY and DM_M in all the environments for each of the four genotypic groups (Figure 5a,b). The Pi-GY values of the semidwarf modern cultivars (both durum and bread wheat) were significantly inferior relative to the landraces, indicating that modern cultivars are more stable and better adapted to varying environments (Figure 5a). Conversely, the Pi-DM_M values of both the bread wheat groups as well as of the durum landraces were inferior to the values of the modern durum cultivars. A significant difference in Pi-DM_M values was observed only between the bread wheat landraces and the modern durum cultivar group (Figure 5b).

The drought susceptibility index (S) was calculated only for the semiarid environments during the 2019 season as the water difference was notable (Figure 6). Three landraces with severe lodging (Juljuli 3.75, Kandaharia 3.75, and Palestinskaya 3.5 lodging rates; Figure 4) were excluded from the S value calculation in order to minimize the confounding effect of yield reduction due to lodging. All landraces, both durum and bread wheat, were more tolerant to drought than modern cultivars. The drought tolerance is reflected in lower drought yield percentage out of the control mean yield (Table 3).

## 3. Discussion

Historically, the Mediterranean basin was the most important area of durum wheat production [18]. According to Zampieri et al. [19], wheat production reliability in the Mediterranean and the Near East will be threatened by climate change with a 1.5 °C global warming as the minimum scenario in the coming decades. Wheat breeding requires reliable indicators to estimate crop resilience under climate change. In this study, field-based evaluation under five environments was performed to evaluate agronomic- and yield-related traits of durum and bread wheat landraces originating from the Near East. The subset of landraces was chosen from the wide and exotic IPLR collection [13] and, as expected, were characterized by phenotypic diversity compared to semidwarf modern cultivars as expressed in the PCA (Figure 1). The Green Revolution breeding has had a clear agronomic imprint on general crop productivity across all traits. Modern cultivars are characterized by higher grain productivity parameters such as GY, HW, and HI, whereas landraces are characterized by late phenology (GDDtH), higher investment in biomass productivity (extensive tillering and PH), and higher GPC (Figure 1). In the driest environment (Gilat19), Lubnani Kisra a bread wheat landrace, had the highest yield among all landraces, 2.6 t ha^−1^, which constituted 80.4% of the modern bread cultivars yield (3.27 t ha^−1^) in that season (Figure 1). The same environment exhibited the highest canopy temperature 87–95 days from emergence (Figure 1). The mixture of bread landraces had the highest number of spikes per square meter in the (L) supplementary irrigation, but this was not translated to yield advantage (2.5 t ha^−1^). This finding is not in accordance with those of a previous study that showed that a modern cultivar mixture had a higher yield than the mean of their pure components [20], especially under low-pesticide cropping systems. In another study, the grain yield advantage of a landraces mixture was only observed in unfavorable environments [21].

Phenotypic variability between genotype groups was subjected to a significant GxE interaction in terms of PH, GN, and GY (Table 1). Earliness is a crucial prerequisite for the adaptation of wheat varieties to the Mediterranean climate [22]. The Mediterranean Basin is the main suitable environment for durum crop [23]; nevertheless, climatic variability occurs across the region, and certain (south-easterly) locations are subjected to frequent drought events, particularly during grain filling [24]. When water availability diminishes and temperatures are high, late grain filling is severely penalized [19]. Earliness seemed to be a crucial factor in promoting yield among landraces (Figure 2) with clear evidence that late -lowering landraces exhibit a high yield penalty due to terminal drought and heat. Similar to the modern cultivars [19], and within the local landraces evaluated in this study, GDDtH was not affected by the environment (Table 2). However, wheat’s vegetative growth period is expected to be shorter by 11 days, as was recently shown for a rain-fed Mediterranean environment [25]. In such scenario, there might be an advantage to sow wheat earlier than mid-November (the common sowing practice) to ensure grain filling is shifted earlier in spring. We hypothesize that the late phenology of the adapted genotypes (e.g., landraces) could be an attractive resource for breeders aiming to adjust the optimal crop phenology and to minimize possible penalties on productivity (assuming sowing early genotypes in October will result in drastic shortening of the growing season and restrict crop development). Jatayev et al. [26] argued that in dryland farming, tall wheat might offer higher yield than Green Revolution cultivars. In our study, modern cultivars maintained significantly higher yield in comparison to landraces across the environmental range (Figure 3a,b). Notably, the most humid environment did not provide any advantage for either modern or landraces groups compared to the rest of the environments. De facto, elevated precipitation in 2020 in Rishon LeZion increased biomass accumulation (Figure 3c) and might have caused severe lodging and consequently yield losses. The genotypes in our study tended to have distinct lodging susceptibilities across environments, excluding Gilat19 (L) (Figure 4). While the modern cultivars scarcely lodged, the landraces lodging rate ranged between moderate to severe. A high plant stature promotes lodging susceptibility, as reflected in the current study (Figure 4b, Table 2), which can be explained by the elongation of internodes, particularly the basal ones [27] in tall genotypes. Although the lodging phenomenon is hard to predict both in space and time as crop growth progresses, lodging risk increases because of the irreversible physical damage to the canopy and the increase in spike leverage. The significance difference between durum and bread wheat landraces relative to their corresponding semi-dwarf modern cultivars are therefore most evident at the latest lodging score. This is again a reflection of crop improvement achieved during the Green Revolution with short stature reducing lodging susceptibility while increasing GY (Table 2).

The measure of a genotype’s general superiority (Pi) for the cultivar–location interaction is calculated as the distance mean squared between the cultivar’s response and the maximum response averaged over all locations [28]. The lower the index value, the more stable the genotype and the better is suited to different environments. Regarding our results, the modern cultivars had an advantage over the landraces for GY (Figure 5a) both for durum and bread wheat. These results are similar to previous study regarding higher GY-Pi for modern cultivars over landraces [5]. In contrast, landraces had higher Pi in DM_M than modern cultivars both for durum and bread wheat (Figure 5b). The high biomass accumulation in stressful environments where lodging risk was reduced (Figure 4) might hint to the suitability of some productive landraces as forage varieties for arid environments. In terms of drought susceptibility, the landraces had a lower fluctuation of GY than modern cultivars (Figure 6). Opposed to the trend that was shown for GY-Pi, landraces had a lower drought susceptibility index (S) than modern cultivars (Figure 6, Table 3). This was first reported by Fischer and Maurer [4], where tall cultivars (of bread wheat and barley) had a lower S value (higher drought resistance) than dwarf bread wheats. The S values for individual genotypes (Figure 6, Table 3) suggest that variation between landraces and modern cultivars is related to phenotypic variance in traits such as earliness, PH, lodging tendency, and GY. The landraces with a lower S also had low yield across the studied environments (Figure 2). As a low S reflects low productivity, a very clear and well-studied trade-off, it might also draw breeders’ attention to landraces with intermediate S values. These show higher drought resistance compared to modern cultivars, but are superior in comparison to the rest of the landraces in important traits such as GY, DtH, and Lg. We identified five landraces with superior performance across all environments for further investigation; Lubnani Kisra, Diar Alla, Hittia Soada, 8238, and Gaza (Table 3). This study mainly focused on the agronomic aspects comparing exotic landraces with highly adapted semi-dwarf varieties. A possible interesting follow-up step may be to explore additional perspectives such as quality attributes of the highly preforming and most-adapted landraces. As can be clearly seen from the current work, wheat landraces were low-yielding across all environments in comparison to the GY of semi=dwarf varieties. However, prices driven by market demands for special heritage wheat grain may compensate farmers’ income and may indicate commercial feasibility. We recommend focusing on some early heading candidates within the panel. A tailor-made agronomic practice should also be studied to optimize their field performance and to encourage their use under the appropriate management.

## 4. Materials and Methods

### 4.1. Germplasm Selection

A representative landraces panel was chosen from the IPLR [13] collection based on documented passport data assuring their authenticity and preliminary phenotypic characterization. The panel was constructed of four main groups: landraces of *T. durum* (*n* = 7), landraces of *T. aestivum* (*n* = 6), modern cultivars of *T. durum* (*n* = 2, cv. C-9, and cv. Solet), and modern cultivars of *T. aestivum* (*n* = 2, cv. Ruta and cv. Gadish). Some of the accessions carry traditional names such as Horani, Nursi, Gaza, and Juljuli (Table A1). The genotypic panel also included two genotypic mixtures of both durum and bread wheat landraces separately, which consisted of the same accessions mixed in equal portions (*n* = 7 durum landrace mixture, *n* = 6 bread wheat landrace mixture).

### 4.2. Field Experiments

The field trials were carried out for two consecutive growing seasons (2018–2019 and 2019–2020) under a semiarid environment (Gilat Experimental Station, located in Negev, south of Israel, 31°33′ N, 34°66′ E) with two levels of supplemental irrigation. In 2019–2020, an additional field experiment was performed under a rain-fed environment (ARO Volcani Institute, Rishon LeZion, 31°98′ N, 34°82′ E). In the semiarid environment (Gilat), a split-plot design was implemented with supplementary irrigation as the main plot and genotype as subplots (18 and 36 m^2^ for 2018–2019 and 2019–2020, respectively) in a randomized block design (*n* = 4). Precipitation in Gilat during the 2018–2019 season was 129 mm. The difference between the supplementary irrigation treatments was Δ105 mm (totaling 262 mm and 367 mm in the L and H water treatments, respectively). In 2019–2020, precipitation in Gilat was 303 mm, and Δ50 mm was the difference between the supplementary irrigation treatments (totaling 363 mm and 413 mm in the L and H water treatments, respectively) (Figure A1). In the Mediterranean environment, a randomized block design layout was implemented with a plot size of 18 m^2^ with total annual precipitation of 704 mm with no supplementary irrigation. An in-field HOBO data logger (Onset HOBO^®^ UA-001-64, Bourne, MA, USA) enabled the sequential monitoring of air temperature during the season (Figure A1). Trials started on 29 November 2018 (fields sown on November 12), 9 December 2019 (fields sown on November 17), 3 December 2019 (fields sown on November 21), and harvested in June 2019 and 2020. All plots were adjusted to the practiced field stand of 220 plants m^−2^. Sowing was performed with a 1.6 m row and a double disk drill (WINTERSTEIGER Plot seed XL, Ried, Austria).

### 4.3. Phenotypic Measurements

Phenotypic characterization at the vegetative stage consisted of field stand (FS) seven days from emergence (DfE); growth habit (GH) (1–5 scale, erect–prostrate, respectively); leaf area index (LAI), expressed as the percentage of green canopy cover at 7, 15, and 30 DfE and was obtained using an image-analysis mobile-phone app (Canopeo app V2.0, Oklahoma State University, Stillwater, OK, USA); days to heading (DtH), expressed as growing degree days to heading (GDDtH). Phenotypic characterization at reproductive stage consisted of flag leaf area (FLA) (measured only in 2018–2019), chlorophyll content of flag leaf (ChC, measured only in 2018–2019 by SPAD 502, Minolta Chlorophyll meter, Tokyo, Japan); Canopy temperature (CT) was measured in 2019 at 87, 112, and 130 DfE (Gilat); in 2020, at 91, 115 and 129 (Gilat); 95, 120, and 137 DfE (Rishon LeZion; IR 200, IR Thermometer, EXTECH Instruments, Nashua, NH, USA); plant height (PH), measured from soil surface up to terminal spikelet excluding awns; spike length (SpL); lodging (Lg) was monitored three times during the vegetative and reproductive stage [in 2019 at 94, 108, and 117 DfE (Gilat); in 2020, at 83, 91 and 97 (Gilat); 76, 82, and 104 DfE (Rishon LeZion)]; grading the severity and the prevalence of lodging between zero (no lodging) to five (complete lodging); dry matter at heading (DM_H) and maturity (DM_M) was determined based on a 0.25 m^2^ random sample with three replications from each subplot. These subsamples at maturity (DM_M) were also used to calculate spike number per meter square (SpN), spike weight per meter square (SpW), grain weight per meter square (GW), grain number per meter square (GN), chaff per meter square (Cf), and harvest index (HI). Following mechanical harvest at maturity, phenotypic characterization of yield-related traits included: grain yield (GY; t ha^−1^;); thousand-kernel weight (TKW; g); test weight (HW; kg hl^−1^); grain protein content (GPC; %).

### 4.4. Statistical Analyses

Descriptive statistics were calculated on the raw dataset to illustrate the variable distribution. To assess the possible effect of genotype, group type (modern and landrace bread and durum wheat), and environment on the different traits, ANOVA was implemented. PCA based on the correlation matrix of genotypes mean was performed jointly for all environments. Initially autocorrelated variables were removed from the analysis. The superiority index was calculated for GY and DM_M according to Lin and Binns [28] by the following equation:$$P_{i}=\frac{{\left(\sum_{j=1}^{n}\left(X_{ij}-M_{j}\right)\right)}^{2}}{2n}$$
where *n* represents the number of environments, *j* represents a specific environment, *X_ij_* represents the mean of a specific wheat line in an environment, and *M_j_* represents the maximum value that was achieved in that environment for the target trait. The drought susceptibility index (S) was calculated for GY according to Fischer and Maurer [4] by the following equation:
$$S=\frac{1-\frac{Y_{D}}{Y_{W}}}{1-\frac{Y_{MD}}{Y_{MW}}}$$
where the parameters *Y_D_* and *Y_W_* represent the yield of the (L) and (H) supplementary irrigation, respectively, for a specific genotype; and the parameters *Y_MD_* and *Y_MW_* represent the yield average in each of the environments. All the statistical analyses were performed using JMP^®^ ver.pro 15.0 statistical package (SAS Institute, Cary, NC, USA).

## Figures and Tables

**Figure 1 plants-10-02612-f001:**
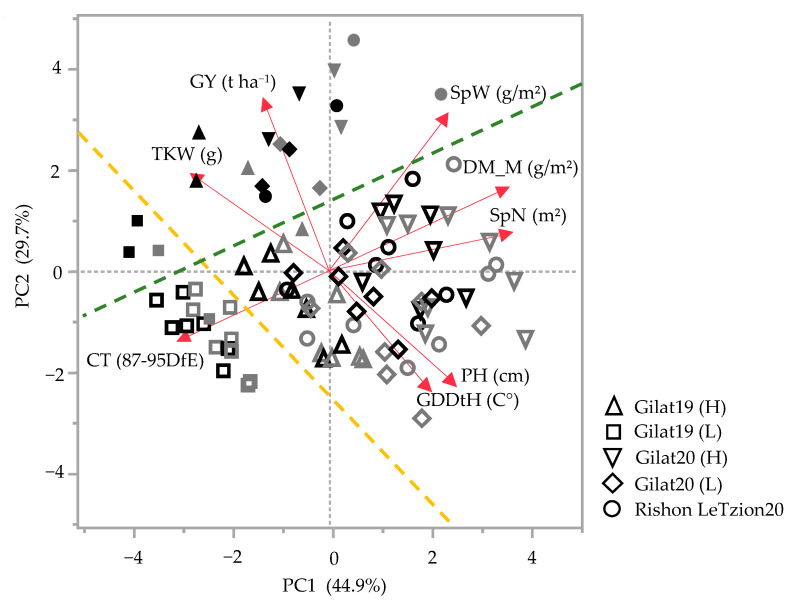
Principal component analysis of agronomic traits under five environments across two seasons, 2018–2019 and 2019–2020. Modern bread wheat cultivars (solid gray symbols), modern durum cultivars (solid black symbols), bread wheat landraces (hollow gray symbols), and durum wheat landraces (hollow black symbols). Biplot vectors are trait factors loading for PC1 and PC2. GY, grain yield; SpW, spike weight; DM_M, dry matter at maturity; SpN, spike number; PH, plant height; GDDtH, growing degree days to heading; CT, canopy temperature; and TKW, thousand kernel weight. Green dashed line separates the modern cultivars from the landraces (with two exceptions). Yellow dashed line separates Gilat19 (L) from the rest of the environments.

**Figure 2 plants-10-02612-f002:**
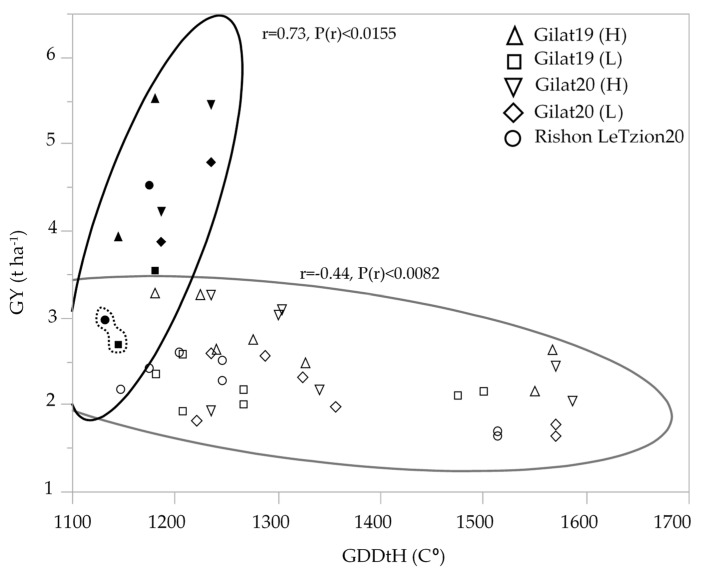
Association between durum wheat grain yield (t ha^−1^) and growing degree days to heading across five environments: modern durum cultivars (solid symbols); durum landraces (hollow symbols). The GY mean of modern cv. Solet under the two most extreme environment is marked by a dashed line.

**Figure 3 plants-10-02612-f003:**
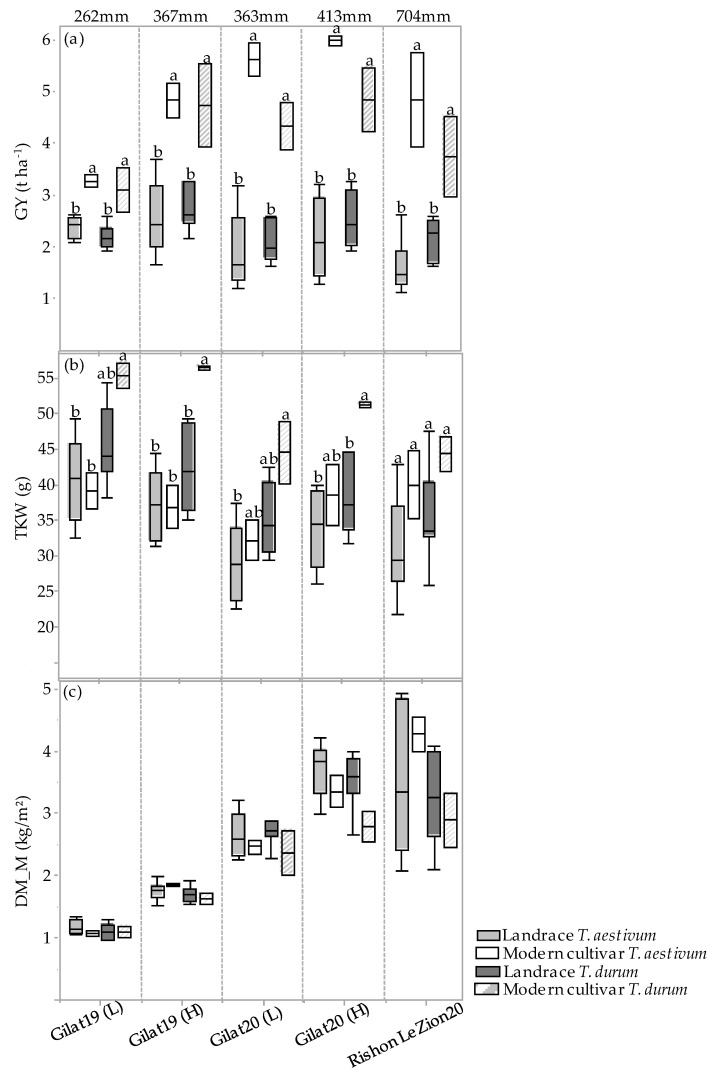
Genotypic groups’ mean comparison under five environments. Grain yield (GY) (**a**), thousand kernel weight (TKW) (**b**), and dry matter accumulation during maturity (DM_M) (**c**). Different letters indicate significant differences between genotypic groups within environments using the Tukey–Kramer test (*p* ≤ 0.05).

**Figure 4 plants-10-02612-f004:**
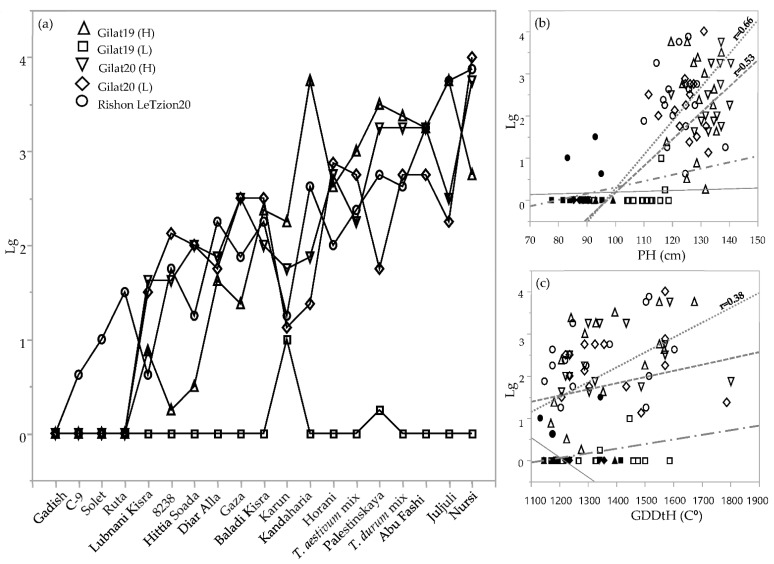
Lodging rate of the IPLR subset panel. Lodging score (from 0 = no lodging to 5 = complete lodging) across 5 environments during 2018–2019 and 2019–3020 seasons (**a**). Correlations between lodging, plant height (**b**), and growing degree days to heading (**c**) of 19 genotypes in 5 different environments. Modern cultivars (solid symbols); landraces (hollow symbols); durum landraces (dotted line); bread wheat landraces (dashed line); durum modern cultivars (continuous line); bread wheat modern cultivars (dotted dashed line). Slopes values are presented only where the linear model was significant (*p* < 0.05).

**Figure 5 plants-10-02612-f005:**
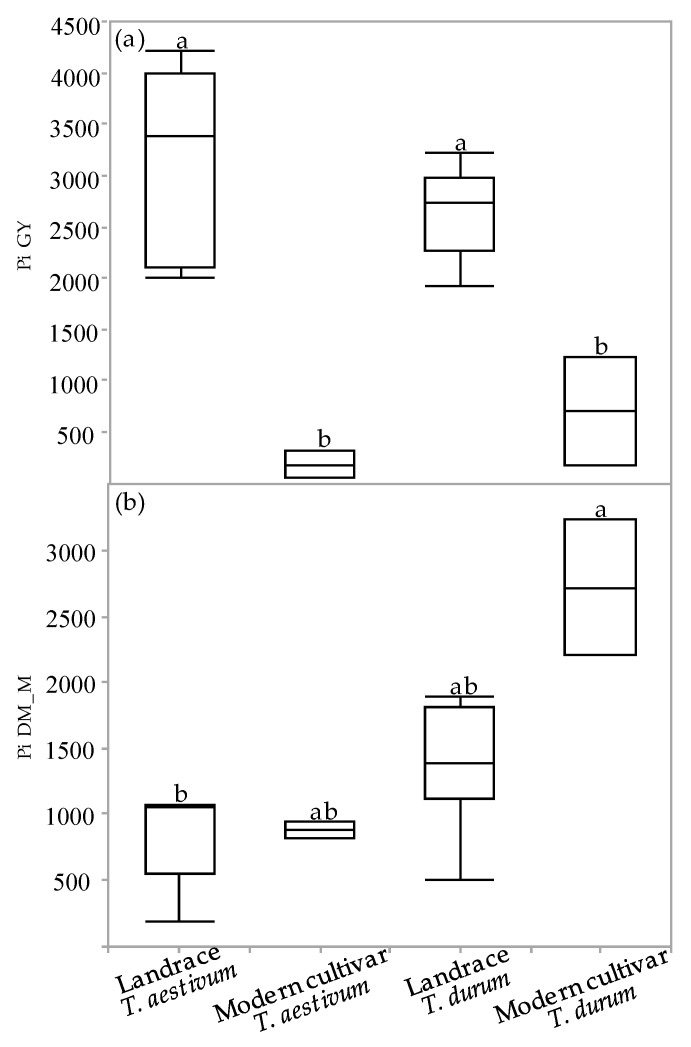
Superiority index (Pi) of genotypic groups across environments: Pi for grain yield (**a**) and dry matter biomass accumulation at maturity (**b**). Different letters indicate significant differences between genotypic groups using Tukey–Kramer’s test (*p* ≤ 0.05).

**Figure 6 plants-10-02612-f006:**
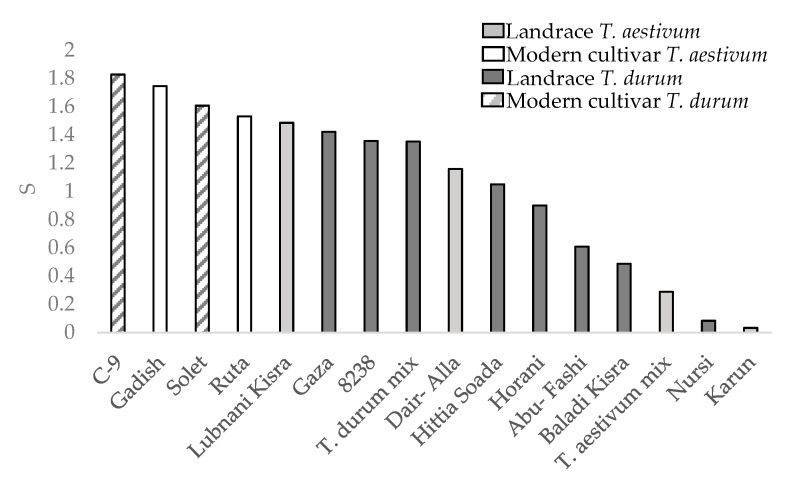
Drought susceptibility index (S) between wheat genotypes.

**Table 1 plants-10-02612-t001:** Analysis of variance for the effects of genotype group (G), environment (E), and their interactions (GXE) for phenotypic traits: growing degree days to heading (GDDtH), canopy temperature (CT), plant height (PH), grain protein content (GPC), thousand kernel weight (TKW), dry matter during maturation (DM_M), spike number (SpN), grain number (GN), and harvest index (HI).

Source of Variance	d.f. ^1^	Mean Square									
		GDDtH	CT	PH	DM_M	SpN	GN	TKW	GY	HI	GPC
		(C°)	(87–95 DfE)	(cm)	(g/m^2^)	(g/m^2^)	(g/m^2^)	(g)	(t ha^−1^)		(%)
G	3	124,218.2	0.8	6585.2	506,785	105,516	364,442,698	703.7	2882.1	0.1	43
		**				***	***	***	***	***	***
E	4	7480.1	91	417.2	13,039,098	276,612	610,606,767	202.4	275.4	0	7.4
			***	***	***	***	***	***	***	***	**
GXE	12	652.4	0.6	64.2	231,034	8399	63,187,797	16.1	98.6	0	0.9
				**			***		***		
R^2^		0.2	0.9	0.9	0.8	0.7	0.8	0.6	0.8	0.8	0.5

^1^ degrees of freedom. **, ***, and n.s. indicate significance at *p* < 0.05 and 0.001 and non-significance, respectively.

**Table 2 plants-10-02612-t002:** Mean lodging (Lg) score of three independent assessments during the vegetative and reproductive stage, plant height (PH), and grain yield (GY).

Group	Mean Lg 76–94 DfE ^1^ ± SE	Mean Lg 82–108 DfE ± SE	Mean Lg 97–117 DfE ± SE	Mean PH (cm) ± SE	Mean GY ± SE
Landraces *T. aestivum*	0.56 ± 1.01 ^a^	0.6 ± 0.8 ^a^	1.81 ± 0.2 ^a^	126 ± 1.7 ^a^	2.2 ± 1.6 ^a^
Modern cultivars *T. aestivum*	0 ^ab^	0.01 ± 0.1 ^b^	0.1625 ± 0.1 ^b^	87.6 ± 3.2 ^b^	4.2 ± 3.2 ^b^
Landraces *T. durum*	0.054 ± 0.19 ^b^	0.74 ± 0.8 ^ab^	1.97 ± 0.2 ^a^	121.98 ± 1.6 ^a^	2.4 ± 1.2 ^a^
Modern cultivars *T. durum*	0 ^ab^	0 ^b^	0.15 ± 0.2 ^b^	90.17 ± 3.2 ^b^	4.9 ± 4.7 ^b^

^1^ Days from emergence. Different letters indicate significant differences between genotypic groups across environments using Tukey–Kramer test (*p* ≤ 0.05).

**Table 3 plants-10-02612-t003:** Mean of grain yield (GY), plant height (PH), lodging (Lg), growing degree days (GDDtH), relative drought yield (RDY), and drought susceptibility index (S) of modern cultivars and selected landraces during the 2018–2019 season.

			GY (t ha^−1^)						
Group	Name	*n*	Gilat19 (L)	Gilat19 (H)	PH (cm)	Lg	GDDtH (C°)	RDY ^a^ (%)	S
Modern cultivars:									
*T. aestivum*	Gadish	4	3.4	5.2	87	0	1199.6	65.7	1.75
	Ruta	4	3.1	4.5	93	0.3	1370.27	69.9	1.53
*T. durum*	C-9	4	3.5	5.5	89	0.13	1201.58	64.1	1.83
	Solet	4	2.7	3.9	86	0.2	1159.23	68.4	1.61
Landraces:									
*T. aestivum*	Lubnani Kisra	4	2.6	3.7	126	0.93	1185.9	70.8	1.49
	Diar Alla	4	2.6	3.3	123	1.5	1320.64	77.2	1.16
*T. durum*	Hittia Soada	4	2.6	3.3	120	1.15	1221.4	79.4	1.05
	8238	4	2	2.7	125	1.13	1276.33	73.3	1.36
	Gaza	4	2.4	3.3	110	1.45	1193.2	72.0	1.42

^a^ Adjusted mean drought yield as a percentage of Gilat19 (H) mean yield.

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
