# Peer review of "In-Field Comparative Study of Landraces vs. Modern Wheat Genotypes under a Mediterranean Climate"

_plants, 2021, doi:10.3390/plants10122612_

Round 1
Reviewer 1 Report
On-field comparative study of landraces vs. modern wheat genotypes under Mediterranean climate
This is very interesting manuscript focusing on the field performance of several landraces from the IPLR collection in comparison to modern wheat cultivars. Worldwide, wheat yield is limited by rainfall and high temperatures. The results showed that water availability had a major impact on crop performance and five landraces were found to have superior performance across different environments. Due to the climate change this founding is of the great importance. Thus, the results are valuable and I think that minor changes must be made before its publishing.
Specific comments are listed below.
Page 1, line 2: GY - please change into the grain yield (GY)
Page 1, line 24: DtH - what is DtH, it is not clear without full explanation in this part of the manuscript
Page 1, lines 37-39: Ben-David et al. [9] suggested that wheat landraces might improve wheat establishment under fluctuation water availability, enhancing grain yield (GY). – The sentence is a little confusing – did you mean wheat landraces with dwarfing genes….as was mentioned in the references?
Page 3, Figure 1: the explanation for TKW represented in the PCA diagram is missing, please add explanation in the figure legend.
Page 3, Table 1: data for SpW and HW are missing
Page 16, line 423: the page number is missing, please chenge into the 1978, 29, 897-912
Page 16, line 429: of northern syria – please use a capital letter
Page 16, line 431: Journal of Animal and Plant Sciences 2011 – please use a journal abbreviation and add volume and page numbers
Page 16, line 435: In; CIHEAM: Bari, 2014; Vol. 110, pp. 147–149. – please delete the semicolon after In; put the year of publishing in the bold, delete Vol.
Page 17, line 437: Agron. J. 1999, 91, 368 – please change into the 368-373
Page 17, line 446: please change the journal data J. Sci. Food Agric. 2020, 100, 4083-4092
Page 17, lines 447-448: please correct the reference Denčić, S., Kastori, R., Kobiljski, B.; Duggan, B. Evaluation of grain yield and its components in wheat cultivars and landraces under near optimal and drought conditions. Euphytica 113, 43–52 (2000). https://doi.org/10.1023/A:1003997700865
Page 17, line 456: please use journal abbreviation (Environ. Res. Lett.)
Author Response
Point 1: This is a very interesting manuscript focusing on the field performance of several landraces from the IPLR collection in comparison to modern wheat cultivars. Worldwide, wheat yield is limited by rainfall and high temperatures. The results showed that water availability had a major impact on crop performance and five landraces were found to have superior performance across different environments. Due to the climate change this founding is of the great importance. Thus, the results are valuable and I think that minor changes must be made before its publishing.
Specific comments are listed below.
Response 1: We are grateful for the positive feedback from the reviewer.
Point 2: Page 1, line 2: GY - please change into the grain yield (GY)
Page 1, line 24: DtH - what is DtH, it is not clear without full explanation in this part of the manuscript
Response 2: Full form of the abbreviation GY and DtH was added.
Point 3: Page 1, lines 37-39: Ben-David et al. [9] suggested that wheat landraces might improve wheat establishment under fluctuation water availability, enhancing grain yield (GY). – The sentence is a little confusing – did you mean wheat landraces with dwarfing genes….as was mentioned in the references?
Response 3: Thank you for the comment. We decided to delete these sentences from the introduction.
Point 4: Page 3, Figure 1: the explanation for TKW represented in the PCA diagram is missing, please add explanation in the figure legend.
Response 4: TKW explanation was added.
Point 5: Page 3, Table 1: data for SpW and HW are missing
Response 5: This was our mistake. We corrected it by removing SpW and HW from Table 1.
Point 6: Page 16, line 423: the page number is missing, please chenge into the 1978, 29, 897-912
Page 16, line 429: of northern syria – please use a capital letter
Page 16, line 431: Journal of Animal and Plant Sciences 2011 – please use a journal abbreviation and add volume and page numbers
Page 16, line 435: In; CIHEAM: Bari, 2014; Vol. 110, pp. 147–149. – please delete the semicolon after In; put the year of publishing in the bold, delete Vol.
Page 17, line 437: Agron. J. 1999, 91, 368 – please change into the 368-373
Page 17, line 446: please change the journal data J. Sci. Food Agric. 2020, 100, 4083-4092
Page 17, lines 447-448: please correct the reference Denčić, S., Kastori, R., Kobiljski, B.; Duggan, B. Evaluation of grain yield and its components in wheat cultivars and landraces under near optimal and drought conditions. Euphytica 113, 43–52 (2000). https://doi.org/10.1023/A:1003997700865
Page 17, line 456: please use journal abbreviation (Environ. Res. Lett.)
Response 6: All references were corrected according to the reviewer suggestion

Reviewer 2 Report
In this manuscript, Franklin et al investigate the potential benefit that durum and bread wheat landraces may provide to breeding programs with respect to warm temperature and drought conditions of Mediterranean climates. The study takes advantage of a collection of Israeli Palestinian landraces and compares their performance to modern cultivars adapted to the local conditions. In general, I found the study to be performed to a good standard, and the conclusions align well with the data presented in the manuscript. The paper is well written, and apart from a few exceptions (see comments below), I found the figures to be well presented.
Most of my concerns are minor and can be corrected by textual changes.
The only scientific concern I have relates to the analysis of drought susceptibility. As part of this analysis, the authors conclude that the landraces have better drought tolerance than the modern cultivars (as determined by S); however, in Figure 6 and Table 3, the authors have included Solet to support this conclusion. Given the data presented in Figure 2, it seems that Solet may be an outlier for grain yield among the modern cultivars - do the authors then think it is fair to include Solet for support of this conclusion? The authors are likely to have an explanation for this point, but I was slightly concerned about this area of the paper.
Following on from this, I think it would be helpful to highlight Solet in Figure 2.
Other minor concerns
- Please define acronyms/abbreviations in the main text where first used, rather than in materials and methods. Examples include "SI", "H" and "L".
- For figures 3 and 5, it would help readers draw a comparison between landraces and modern cultivars more easily if the boxes are arranged such that the landrace T. aestivum is alongside the modern cultivar T. aestivum, and the same for durum, rather than the current arrangement.
- L137 - correct to "might have also resulted"
- L252-255 - typographical errors to correct "recently sown" should be "recently shown"(?), "sow wheat early" should be "sow wheat earlier", "hypotise" should be "hypothesize".
Author Response
In response to the comments of reviewer #2:
Point 1: In this manuscript, Franklin et al investigate the potential benefit that durum and bread wheat landraces may provide to breeding programs with respect to warm temperature and drought conditions of Mediterranean climates. The study takes advantage of a collection of Israeli Palestinian landraces and compares their performance to modern cultivars adapted to the local conditions. In general, I found the study to be performed to a good standard, and the conclusions align well with the data presented in the manuscript. The paper is well written, and apart from a few exceptions (see comments below), I found the figures to be well presented.
Most of my concerns are minor and can be corrected by textual changes.
Response 1: We highly appreciate the reviewer insights and comments.
Point 2: The only scientific concern I have relates to the analysis of drought susceptibility. As part of this analysis, the authors conclude that the landraces have better drought tolerance than the modern cultivars (as determined by S); however, in Figure 6 and Table 3, the authors have included Solet to support this conclusion. Given the data presented in Figure 2, it seems that Solet may be an outlier for grain yield among the modern cultivars - do the authors then think it is fair to include Solet for support of this conclusion? The authors are likely to have an explanation for this point, but I was slightly concerned about this area of the paper.
Response 2: Modern cv. ‘Solet’ lower yield at the two most extreme environments is indeed an outlier compared to the rest of the modern cultivars mean GY in this experiment. Nevertheless, its drought susceptibility (Table 3) was high as the rest of the modern cultivars. Figure 2 emphasizes not only the advantage of modern durum cultivars in GY but also their earliness (DtH) across all field trails compared to durum landraces. The relative low yield of ‘Solet’ was reflected in the most two extreme environments (wettest and driest environment). In the other three environments, ‘Solet’ maintained higher grain yield compared to durum landraces (Figure 2) and together with cv. ‘C-9’ the leading durum variety in Israel, fairly represent the commercial elite durum germplasm currently available for farmers.
Point 3: Following on from this, I think it would be helpful to highlight Solet in Figure 2.
Response 3: We followed the reviewer comment and highlighted Solet GY means under the two extreme environments in Figure 2.
Point 4: Other minor concerns
Please define acronyms/abbreviations in the main text where first used, rather than in materials and methods. Examples include "SI", "H" and "L".
Response 4: Full form of the abbreviation was added.
Point 5: For figures 3 and 5, it would help readers draw a comparison between landraces and modern cultivars more easily if the boxes are arranged such that the landrace T. aestivum is alongside the modern cultivar T. aestivum, and the same for durum, rather than the current arrangement.
Response 5: Both figures 3 and 5 were edited following the reviewer suggestion.
Point 6: L137 - correct to "might have also resulted"
L252-255 - typographical errors to correct "recently sown" should be "recently shown"(?), "sow wheat early" should be "sow wheat earlier", "hypotise" should be "hypothesize".
Response 6: Spelling mistakes were corrected following the reviewer remarks.

Reviewer 3 Report
1). Manuscript ID: Plants-1466144
2). Manuscript Title: On-field comparative study of landraces vs. modern wheat genotypes under Mediterranean climate
3). Comments:
The research described in this article was carried out in an efficient manner. However, the following minor corrections should be done before acceptance.
--Moderate English changes required.
--Add scientific authority at the end of binomial names of all species when they are mentioned for the first time in the manuscript.
--Include full forms of all abbreviations/acronyms mentioned in the manuscript.
Author Response
In response to the comments of reviewer #3:
Point 1: The research described in this article was carried out in an efficient manner. However, the following minor corrections should be done before acceptance.
--Moderate English changes required.
Response 1: We reviewed the manuscript and corrected moderate English spelling.
Point 2: --Add scientific authority at the end of binomial names of all species when they are mentioned for the first time in the manuscript.
Response 2: The scientific authority at the end of binominal manes were added to T. durum and T. aestivum in lines 54 and 66 respectively.
Point 3: --Include full forms of all abbreviations/acronyms mentioned in the manuscript.
Response 3: Full form of the abbreviation was added.
